# Efficient preservation and breakup of liquid sheets using screen-projected particles

**Jong-Hyun Kim**[1], **Jung Lee**[2]*

**1** Department of Software Application, Kangnam University, Yongin, Gyeonggi, Republic of Korea,
**2** Department of Convergence Software, Hallym University, Chuncheon, Gangwon, Republic of Korea

* airjung@hallym.ac.kr

**Data Availability Statement:** All relevant data are within the manuscript and attached video file.

**Funding:** This work was supported in part by the Basic Science Research Program through the National Research Foundation of Korea (NRF)

## Abstract

In this paper, we propose a technique that can efficiently express the preservation and breakup of liquid sheets by eliminating over-preserved liquid sheets using the motion of water particles projected onto the screen. First, we project three-dimensional water particles onto a two-dimensional screen. When multiple particles are projected onto the same pixel, we select one of the front most particles as a screened particle by comparing their depth values. Based on the anisotropic kernel and density, the motion of the screened water particles is tracked to determine whether preservation and breakup should be performed. As a result, new water particles are added or existing ones are deleted, which makes it possible to express two characteristics of particle-based fluids: preservation and breakup of liquid sheets. The proposed technique is based on particle-based fluids, which can be used to remove the over-preserved liquid sheets and thus improve the quality of liquid sheets without surface noise.

## Introduction

Fluid simulations are used in various fields such as film, game, VFX (visual effects) and CFD (computational fluid dynamics). In particular, analyzing and simulating the motion of a fluid is a necessary process to capture the detailed characteristics of the fluid. It is commonly required for expressing the water, fire, and smoke. As is well known, the scale and detail of the simulations depend on the grid resolution used, and it is difficult to express the details of the fluid above a fixed resolution. The upsampling technique computes fluid simulations in a low-resolution grid space and then applies the high quality turbulence model to express fluid details [1–3]. This approach can represent high quality turbulence while requiring low computational complexity based on a low-resolution grid. Particularly, in the field of water simulations, researches are being made to express noise-free, smooth and detailed fluid surfaces such as liquid sheets [4, 5], thin surface features [6, 7], and capillary waves [8].

When reconstructing fluid surfaces using water particles, blobs and holes are created on the surfaces depending on the distribution of particles. Various anisotropic approaches have been proposed to reduce this problem. However, techniques for reliably expressing the preservation and breakup of liquid sheets have not been presented so far. Several methods of representing

funded by the Ministry of Science, ICT & Future Planning (2017R1C1B5074984), and in part by a Hallym University Research Fund (HRF-201909-018).

**Competing interests:** The authors have declared that no competing interests exist.

liquid sheets in particle-based fluids have been proposed, but most of them have introduced an approach to filling holes on the fluid surface when water particles are not evenly distributed [4, 5]. This method analyzes the movement of adjacent particles to predict where the hole will occur and adds new water particles to this area, so the liquid sheets seem to be well preserved. However, the newly added particles cause the surface to over-preserve and appear as complex noise. In this paper, we describe the problems in previous particle-based liquid simulation techniques that represent liquid sheets and show how to solve them.

## Problem statement

Particle-based fluid simulation is often used to represent water effects in a game or VFX. Among them, FLIP (Fluid-Implicit Particle) solver is used for expressing various materials (*granular*, *water*, *smoke*, *lava*, *elastoplastic*, etc.) because it can express high quality fluid motion [3, 9]. Naive FLIP-based water simulation, however, can sometimes cause particles to scatter like splashes or fail to properly capture the characteristics of liquid sheets (see Fig 1-left).

Several studies have been proposed to solve this problem, most of which add new water particles to the holes that occur during the reconstruction of fluid surfaces [4, 5]. These methods are intuitive, but since they can not express the breakup effects of liquid sheets, their results show noisy and viscous sheets. (see Fig 1-right). It is essential to consider the preservation and breakup of liquid sheets to express them clearly. In this paper, we propose a novel framework for expressing detailed liquid sheets by solving this problem.

## Related work

The level-set technique proposed by Osher and Sethian is one of the widely used methods for physically based simulations and CFD [10]. It can be used to calculate accurate implicit function in surface tracking process to extract zero-contour of liquid surface [11] or to reduce numerical dissipation through polynomial level-set advection in fluid advection process [12]. This approach has been improved and cited in a number of subsequent studies because it allows for easy representation of complex fluid surface topologies by using the feature that level-set values are zero near fluid surfaces [13–15]. Heo and Ko proposed a technique for expressing thin surfaces of fluids with a spectrally refined level-set technique and a level-set initialization technique with high-order accuracy [16]. However, this method was not sufficient to represent the characteristics of thin fluid surfaces tearing or breakup. Mesh-based surface tracking methods have been proposed to solve this problem, but they have made the algorithm more complex because of the complex topology changes [17–19].

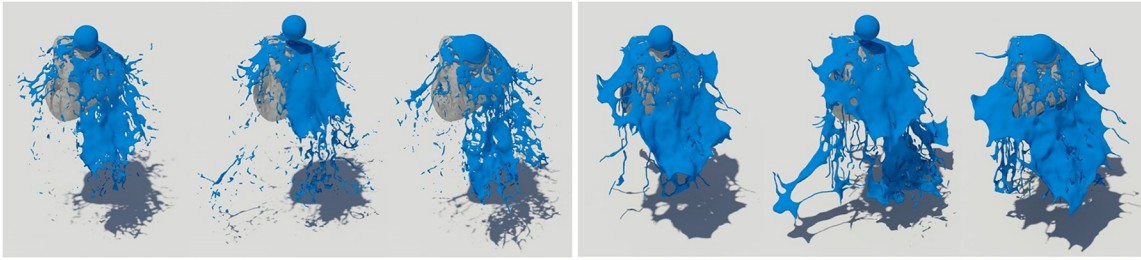

**Fig 1. Problem statement: Over preserved liquid sheets for filling holes in particle-based water simulation with previous methods (left: Holes in Naive FLIP simulations [21], right: Over preserved liquid sheets [4, 5]).**

In particle-based fluid simulation, various methods have been proposed for reconstructing fluid surfaces from a large number of particles [20–23]. Since particles have no connectivity information, fluid surfaces are reconstructed by approximating the level-set using the density calculated from adjacent particles. However, this approach is prone to producing low-quality fluid surfaces that contain blobs and noise. Particularly in areas where the number of particles is small, the fluid surfaces are not properly represented and holes are created. To solve this problem, Ando and Tsuruno proposed a method for creating and preserving liquid sheets in FLIP-based fluid simulations [4]. This method predicts where holes will be created on fluid surfaces and preserve liquid sheets by adding new particles to fill the holes. They have improved the efficiency of this technique and have also proposed a method to preserve liquid sheets in consideration of the adaptive and anisotropic characteristics [5]. While these approaches are excellent for liquid sheets, the liquid sheets are over-preserved during the process of splitting one fluid particle into two, resulting in noisy fluid surfaces (see Fig 1-right). Recently, research has also been carried out to restore turbulence flow lost by preserving liquid sheets by redistributing the mass according to the number of fluid particles [24]. In addition, techniques have been proposed that synthesize fluids using machine learning techniques to reduce computation time. However, since most of these techniques focus on improving efficiency, it is difficult to express high-quality liquid sheets [25–29].

Studies have also been carried out to describe the discontinuous state of fluid as well as liquid thin sheets. Many methods based on the GFM (Ghost fluid method) have been studied to accurately calculate the discontinuous state occurring near the interface where the liquid is in contact with air or solid [30–35]. Among them, Hong et al. represented the discontinuous fluid by improving the subgrid detail of the fluid simulations by adding a GFM-based jump condition to the surface tension and viscosity at the interface between liquid and solid [35]. However, since this technique is based on the Eulerian approach, it is difficult to integrate it into a particle-based simulation techniques.

## Proposed framework

Our method uses the FLIP algorithm to express the water volume and advect water particles [21]. This method improves the accuracy of advection and is often used to represent large scale water volumes. FLIP is capable of producing incompressible and turbulent flow motion that yields visually convincing behavior. We found this method to be a highly suitable basis for computing smooth flow fields that lead to the development of liquid sheets effects.

Liquid sheets are often observed in phenomena such as water splashes, where thin sheets are preserved or torn as fluid moves. We anisotropically analyze the direction of motion of the fluid in three dimensions based on water particles. In order to efficiently calculate this process in a simulation with a large number of particles, we analyze the motion using screened water particles obtained by projecting 3D water particles onto 2D screen space. Our framework improves the details of fluid surfaces by preserving or discontinuously breaking liquid sheets through the analysis of these particles. A summary of the proposed algorithm is provided below:

1. Calculate the density of the water particles and advect the particles using the FLIP solver.

2. Project 3D water particles onto the screen space using a projection matrix and extract the screened water particles.

3. Extract thin particles from screened water particles using SVD (Singular value decomposition).

4. Extract thin particle pairs between candidate positions using the position, relative velocity, and mass change of water particles.

5. Express the preservation and breakup of liquid sheets by adding or removing water particles using thin particle pairs.

6. Water surfaces are reconstructed from water particles.

### Projection of water particles

Our approach is similar to approaches based pm 2D projective space [36–38]. Fluid motion is approximated by a linear combination of projection functions applied to projected water particles. This projection function encapsulates the properties of the particle, including position and velocity, which are projected onto screen space. We first project each 3D water particle onto screen space and create its projection function by local interpolation of the particle properties within its radius. Using the interpolated depth values in 2D projective space, it becomes possible to calculate the motion of the 3D water particle based on the Navier-Stokes equation in 2D.

### Selection of screened water particles

We use the method proposed by Müller et al. [39] to calculate the depth map from the water particles and extract the screened particles. $W$ and $H$ are the width and height of the screen space, respectively, expressed in pixels. $N_x \times N_y$ represents the resolution of a regular grid divided by a projective spacing $h$, and $r$ is the radius of a 3D water particle. The depth $z_{i,j}$ of each 3D water particle are stored in the depth map $\mathbf{Z} \in \mathbb{R}^{N_x \times N_y}$ at a location of its projected coordinates (see Fig 2).

A point $\mathbf{x}$ in 3D space, $[x, y, z, 1]^T$, is transformed into 2D projective space using the projection matrix $\mathbf{P}$.

$$\begin{bmatrix} x' \\ y' \\ z' \\ w \end{bmatrix} = \mathbf{P} \begin{bmatrix} x \\ y \\ z \\ 1 \end{bmatrix}. \tag{1}$$

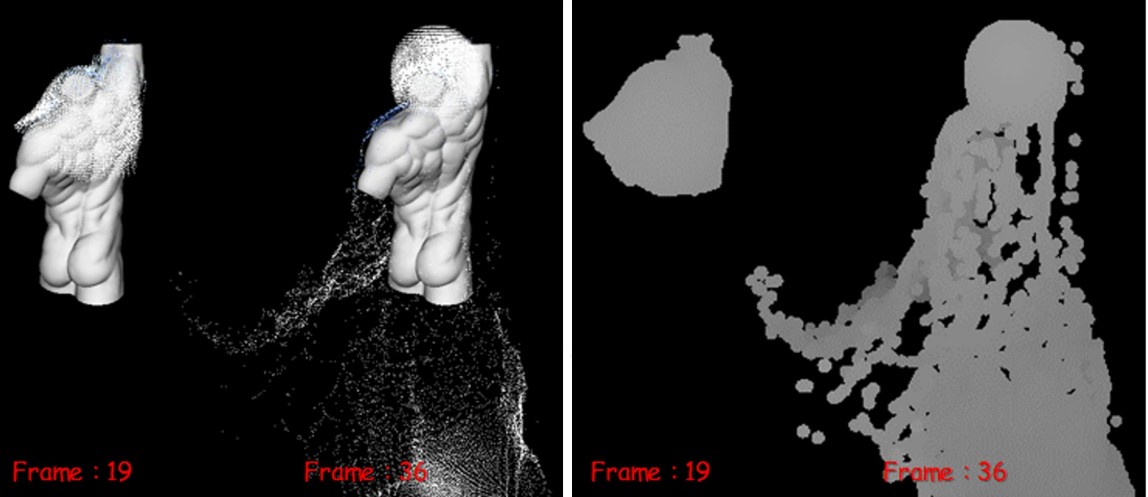

**Fig 2. Visualizing the depth of water particles in projective space (left: Particle view, right: Depth map).**

To avoid distorting the $z$ values during projection, the perspective division is applied only to $x$ and $y$, not to $z$ [39]. Using this scheme, the projected coordinates $(x_p, y_p)$, $z_p$ of each 3D water particle are computed as follows:

$$\underbrace{\begin{bmatrix} x_p \\ y_p \\ z_p \end{bmatrix} = \begin{bmatrix} W \cdot \left( \frac{1}{2} + \frac{1}{2} x'/w \right) \\ H \cdot \left( \frac{1}{2} + \frac{1}{2} y'/w \right) \\ z' \end{bmatrix}}_{\textit{projected coordinates and depth}}, \tag{2}$$

The 3D water particle radius $r$ is projected as follows:

$$\begin{bmatrix} r_x \\ r_y \\ r_z \end{bmatrix} = \begin{bmatrix} rW \sqrt{p_{1,1}^2 + p_{1,2}^2 + p_{1,3}^2}/w \\ rH \sqrt{p_{2,1}^2 + p_{2,2}^2 + p_{2,3}^2}/w \\ r \sqrt{p_{3,1}^2 + p_{3,2}^2 + p_{3,3}^2} \end{bmatrix}, \tag{3}$$

where the term $p_{i,j}$ are entries in the projection matrix $\mathbf{P}$. A single isotropic projected radius is obtained by assigning the same value to radius $r_p$ in screen space, $r_x$, and $r_y$. The particle has three-dimensional coordinates of $x$, $y$, $z$, and we do the same for radius as we compute the projected screen coordinates $x_p$, $y_p$, $z_p$ by applying Eq 7 to these three-dimensional coordinates. In 3D, the radius of the $X$, $Y$, and $Z$ axes is the same, and since $r_x$ and $r_y$ have the same value after the projection, any of $r_x$ and $r_y$ can be used as the value of $r_p$.

Because multiple 3D water particles can be projected into a single entry in the depth map, values ($z_{i,j}$) of depth and index($p_n^{screened}$) of screened water particles are updated using the following equation:

$$z_{i,j} \leftarrow \min(z_{i,j}, z_p - r_z h_{i,j}), p_n^{screened} \leftarrow \text{argmin}(z_{i,j}), \tag{4}$$

where

$$h_{i,j} = \sqrt{1 - \frac{(ih - x_p)^2 + (jh - y_p)^2}{r_p^2}}. \tag{5}$$

Condition $(ih - x_p)^2 + (jh - y_p)^2 \leq r_p^2$ is checked to determine whether the projected coordinate is affected by each node of projective space. If the depth value of the projected coordinate $z_p - r_z h_{i,j}$ is less than $z_{i,j}$, Eq 4 updates the depth $z_{i,j}$ and the corresponding entries with the index of screened water particles $p_n^{screened}$.

Fig 2 shows the depth map extracted from the water particles and the depth value is calculated using the radius $r_p$ of the water particles as described before. Based on the calculated depth value, we identify the 3D water particles closest to the camera, and the identified particles are called screened water particles in this paper.

Fig 3 shows the extracted screened water particles and shows them with the back-face particles classified in a third viewpoint different from the actual projection screen, where back-face particles are particles that are not visible from the camera. In this paper, liquid sheets are expressed using only screened particles in order to improve algorithm efficiency of particle based simulation with high computational complexity.

We have tested whether the screened water particles can be robustly extracted from the static scene of a fixed viewpoint as well as a dynamic scene where the camera rotates every

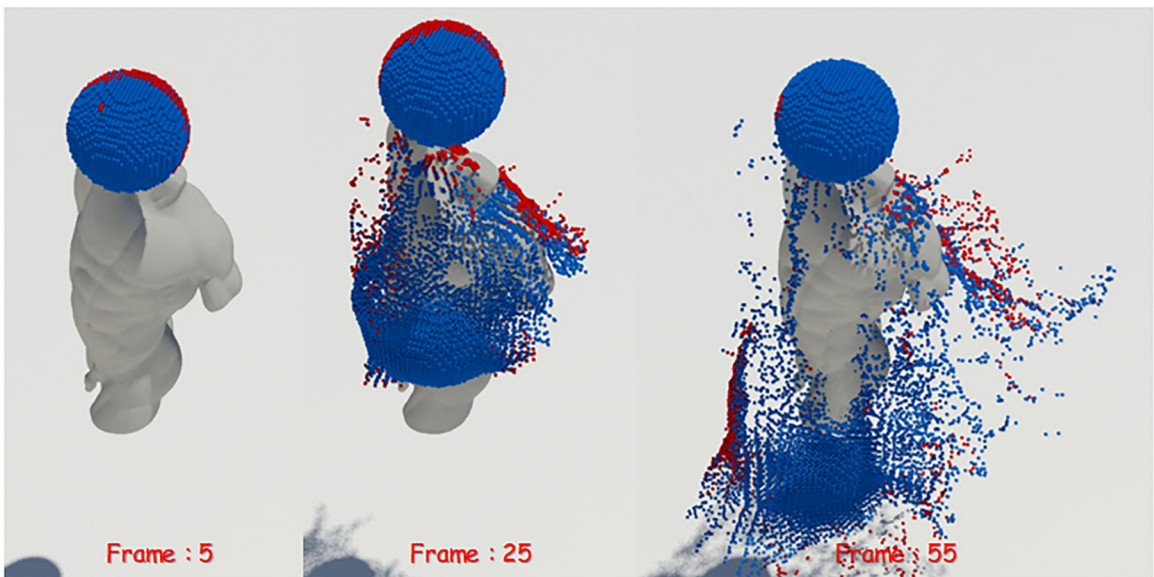

**Fig 3. Screened water particles in rotating camera scene viewed from the third viewpoint (blue: Screened water particle, red: Back-face particle).**

frame. The results will be described later in the results section, where the proposed technique reliably extracts screened water particles even in environments where the camera is constantly rotated (see Fig 4).

## Extraction of thin water particles from screened water particles

In the previous section, we discussed how to use screened water particles for computational efficiency. This section describes how to express the preservation and breakup of liquid sheets to improve the quality of fluid surfaces.

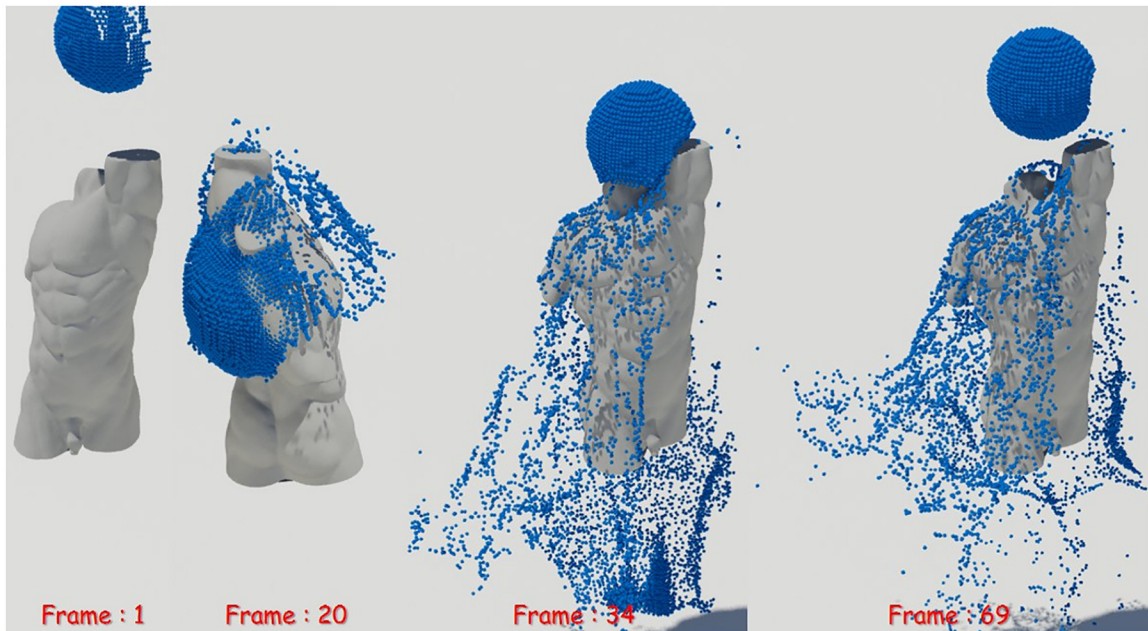

**Fig 4. Screened water particles in rotating camera scene (blue: Screened water particle).**

In this study, the preservation of liquid sheets is modeled based on Ando's method [4, 5] and thin water particles are extracted. Liquid sheets are expressed in the thinly spreading fluid surfaces. We use this anisotropic kernel to find this region and analyze the distribution of adjacent water particles. We also calculate the orientation of each water particle with a weighted average-based covariance matrix $C_i$ (see Eq 6). In this process, we use only screened water particles, not whole particles of the fluid body, as in previous studies.

$$C_i = \frac{\sum_j (p_j - \bar{p}_i) \sum_j (p_j - \bar{p}_i)^T W_{smooth}(p_j - \bar{p}_i, \alpha_1 d_0)}{\sum_j W_{smooth}(p_j - \bar{p}_i, \alpha_1 d_0)}, \tag{6}$$

where $d_0$ represents the initial spacing between water particles, and $\alpha_1$ is the weight value for $d_0$. $\bar{p}_i$ is the position using Laplacian smoothing, and $W_{smooth}$ is the isotropic weighted kernel (see Eqs 7 and 8).

$$\bar{p}_i = \frac{\sum_j p_j W_{smooth}(p_j - p_i, \alpha_1 d_0)}{\sum_j W_{smooth}(p_j - p_i, \alpha_1 d_0)}, \tag{7}$$

$$W_{smooth}(\mathbf{r}, h) = \begin{cases} 1 - \| \mathbf{r} \|^2 / h^2, & 0 \leq \| \mathbf{r} \| \leq h, \\ 0, & \text{otherwise.} \end{cases} \tag{8}$$

The covariance matrix $C_i$ computed by the above equation is used to obtain eigen vectors and eigenvalues using SVD as follows. This is used to calculate the stretching and orientation between adjacent water particles:

$$C_i = \begin{bmatrix} e_1 & e_2 & e_3 \end{bmatrix} \begin{bmatrix} \sigma_1 & & \\ & \sigma_2 & \\ & & \sigma_3 \end{bmatrix} \begin{bmatrix} e_1^T \\ e_2^T \\ e_3^T \end{bmatrix}, \tag{9}$$

where $e_n$ represents the principle axes ordered by variance and $\sigma_n$ represents the size of the stretch. To extract thin particles from water particles, the following conditions were checked: $\sigma_3 \leq \alpha_2 \sigma_1$, where $\alpha_2$ is a threshold that determines the size of the thickness. Fig 5 shows thin particles extracted based on an anisotropic kernel. Observing the movement of the water particles, we can see that the thin particles are extracted after they collide with the solid as they spread out. That is, thin particles are not extracted until water particles collide with the solid, and then begin to be extracted as the fluid surfaces are spread out thinly after the collision.

## Extraction of candidate particles

The extracted thin particles are used to determine the positions of new particles needed to fill the hole. In this paper, we find $(i, j)$, which is a pair of connected particles to fill a hole in a fluid surface. The center of each pair, $\frac{p_i + p_j}{2}$, is stored as a candidate position to be split later, and we choose a pair that satisfies all of the following conditions:

$$\alpha_3 d_0 \leq \| p_i - p_j \| \leq \alpha_4 d_0, \tag{10a}$$

$$\sum_k W_{smooth}((p_i + p_j)/2 - p_k, \alpha_3 d_0) = 0, \tag{10b}$$

$$(p_i - p_j) \cdot (u_i - u_j) > 0, \tag{10c}$$

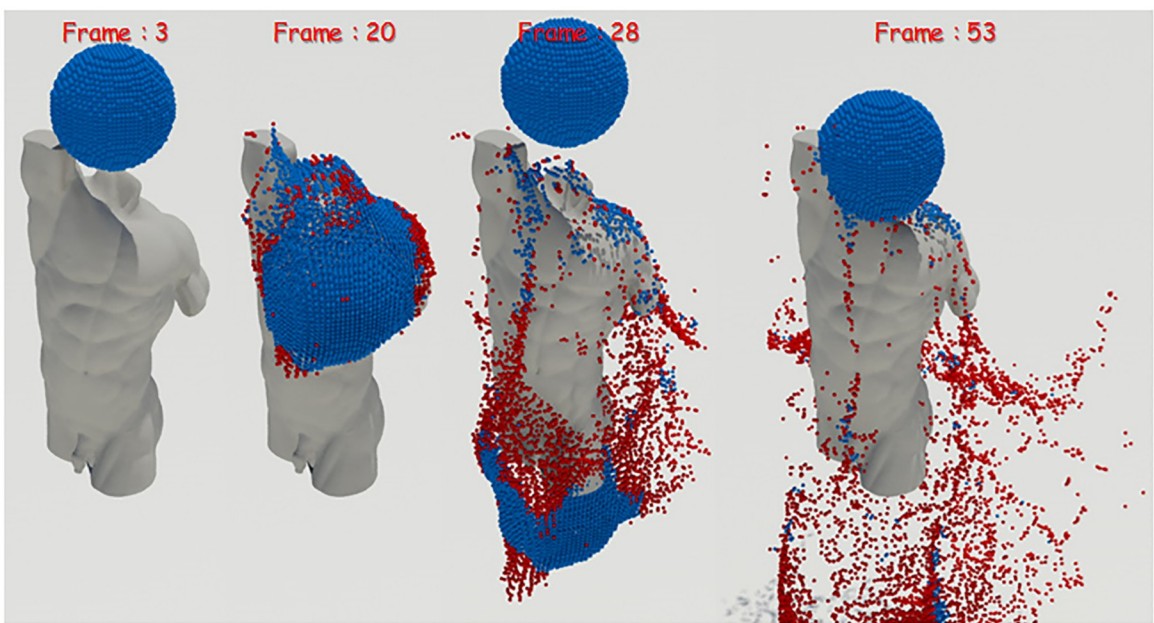

**Fig 5. Thin particles extracted from water particles (Thin particles are colored red and lie on the outer edge of the fluid body).**

where $\alpha_3$ and $\alpha_4$ are the minimum and maximum distances between the candidate particles, respectively, and $u$ is the velocity of the particle. Eq 10a determines whether a hole exists based on the distance between two particles. Eq 10b tests whether there are no water particles within the radius $\alpha_3$ centered around the midpoint of the candidate particles and is a sufficiently thin region. Eq 10c tests whether the distance between water particles increases over time and determines whether the hole is getting bigger. These equations locate candidate particles from previously extracted thin particles, resulting in over-preserved liquid sheets because they extract complicated and noisy fluid surfaces (see Fig 1). In this study, we add the following condition to breakup liquid sheets.

$$\frac{\rho_{t+\triangle t}^{p*} + \rho_t^{p*}}{2} \leq \tau, \tag{11}$$

where $\rho$ is the density of the particles, and $p^*$ is the $(i, j)$ particles that make up the pair. Eq 11 is used to remove the particles from the list of thin particles if the average density of a pair of particles is less than the user defined threshold, $\tau$.

Fig 6 shows the visualization of the mass of water particles. The closer the particle color is from red to blue, the closer its mass value is to zero. Eq 11 suggests a condition for breakup of over preserved liquid sheets, which is used to break down the area of density which is becoming smaller due to the smaller mass, which is used to break up liquid sheets in this study. To summarize, in this study, we proposed a condition to break a pair of water particles in order to express the natural tearing of the fluid surfaces with small density. The intermediate positions of the pairs satisfying all the above conditions are stored in the **S** without duplication.

Fig 7 shows extracted candidate particles. Fig 7-left shows the candidate particles extracted through Eq 10. As the figure shows, when the liquid sheets start to be preserved, the candidate particles are extracted without breakup. After the collision of fluid and solid, candidate particles are extracted continuously. On the other hand, the addition of Eq 11 gives a good representation of the preservation and breakup of thin liquid surfaces as well as liquid sheets. The

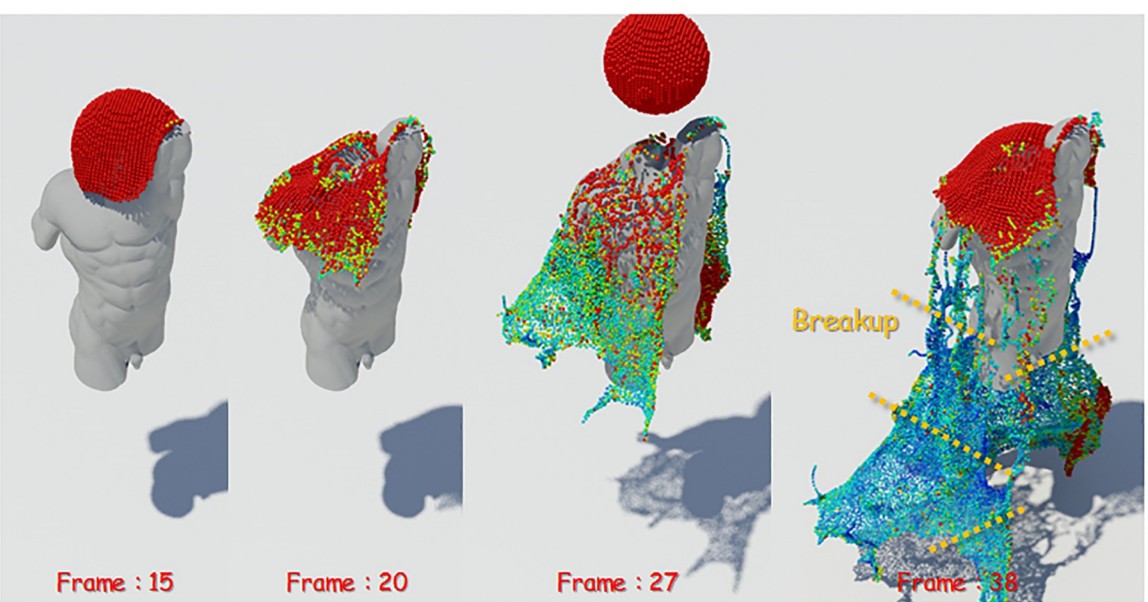

**Fig 6. Mass visualization of thin particles (red: High mass, blue: Low mass, orange dotted line: Breakup posotion of liquid sheets).**

results are well seen in the extraction of candidate particles (see Fig 7-right). Compared with Fig 7-left, Fig 7 shows clearly that the number of candidate particles gradually decreases as the liquid sheets break at the torn surface because the density becomes very small after the collision of fluid and solid.

## Insert and remove water particles

It is not appropriate to add new water particles where the candidate particles are densely arranged. This section describes how to filter unnecessary particles and preserve liquid sheets by adding a minimum number of water particles. $\mathbf{I}$ is a list of finally inserted candidates

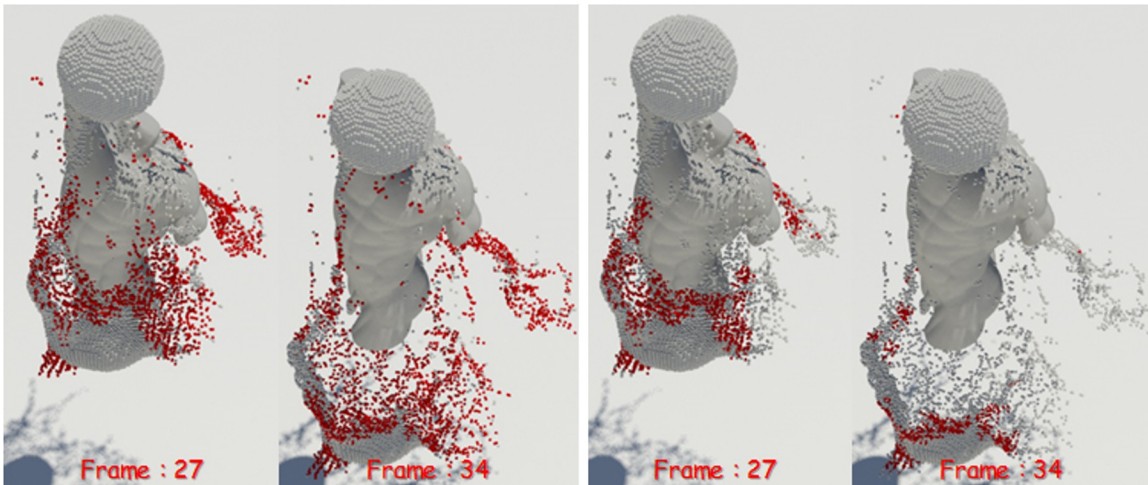

**Fig 7. Candidate particles extraction (red: Candidate particle, left: Only Eq 10, right: With Eqs 10 and 11).**

indices. The first particle $I_1$ is the most sparse candidate within $\mathbf{S}$, as follows:

$$I_1 = \operatorname*{argmin}_{j \in \text{sizeS}} \rho(\mathbf{s}_j), \tag{12}$$

where $\rho_j$ is the density of the $j$th particle. After finding $I_1$ from $\mathbf{S}$, we delete all candidate particles within the radius $\alpha_3 \, d_0$ around $\mathbf{s}_{I_1}$'s surrounding particles. We then search for particles outside the radius $\alpha_3$ around $\mathbf{s}_{I_1}$'s surrounding particles and add them to $\mathbf{N}_{I_1}$. Finally, we find the closest particle $\mathbf{s}_j$ in the $\mathbf{N}_{I_1}$ and add it to $I_2$ (see Eq 13).

$$I_2 = \operatorname*{argmin}_{j \in \mathbf{N}_{I_1}} \| \, \mathbf{s}_j - \mathbf{s}_{I_1} \, \| . \tag{13}$$

The above equation for searching for particles can be calculated as: $I_{n+1} = \text{search}(I_n)$. This means that $I_2$ can be calculated in the same way. Likewise, $I_3, I_4, \ldots, I_{n+1}$ can also be solved by iterative calculations. If there is no candidate position that satisfies the search condition, Eq 12 is used instead.

After the water particle is split into two particles, its properties are calculated by linear interpolation, and the velocity $u$ and particle density $\rho$ mapped to the grid are calculated as follows:

$$u(\mathbf{x}) = \frac{\sum_i m_i \mathbf{u}_i W_{sharp}(\mathbf{p}_i - \mathbf{x}, \alpha_5 d_0)}{\sum_i m_i W_{sharp}(\mathbf{p}_i - \mathbf{x}, \alpha_5 d_0)}, \tag{14}$$

$$\rho(\mathbf{x}) = \sum_i m_i W_{smooth}(\mathbf{p}_i - \mathbf{x}, \alpha_1 d_0), \tag{15}$$

where $\alpha_5$ is the weight value for velocity $u$, and $W_{sharp}$ is the kernel function:

$$W_{sharp}(\mathbf{r}, h) = \begin{cases} h^2 / \| \, \mathbf{r} \, \|^2 - 1, & 0 \leq \| \, \mathbf{r} \, \| \leq h, \\ 0, & \text{otherwise.} \end{cases} \tag{16}$$

If one of the following conditions is satisfied, the water particle $i$ is removed (see Eqs 17 and 18):

$$\rho_i > \alpha_6 \rho_0 \ and \ \sigma_3 \geq \alpha_2 \sigma_1, \tag{17}$$

$$\| \, \mathbf{p}_i - \mathbf{p}_j \, \| < \alpha_7 d_0, \quad \text{for any } j \neq i, \tag{18}$$

where $\alpha_6$ is the maximum density value and $\alpha_7$ is the minimum distance between the water particles. The water particles satisfying Equations A and B are removed and their masses are returned back to the original fluid particle before they are split to preserve the total mass. The process of adding and deleting water particles has a serial structure where $I_n$ is processed and then $I_{n+1}$ is processed. So we used the same method as Ando et al.'s algorithm [4, 5].

## Surface reconstruction

When restoring the fluid surface, we use the SVD value used to extract the thin particles. In this paper, we reconstruct the fluid surfaces in detail by reusing the anisotropic properties of SVD with the algorithm proposed by Yu et al. [23]. The level-set of particles is calculated as follows (see Eq 19).

$$\phi(\mathbf{x}) = \min_i (\| \, G_i(\mathbf{p}_i - \mathbf{x}) \, \|) \tag{19}$$

where $G_i$ is the transformation matrix for the water particle $i$ and its properties are as follows

(see Eq 20).

$$G_i = \frac{1}{k_s} \begin{bmatrix} e_1^T \\ e_2^T \\ e_3^T \end{bmatrix}^T \begin{bmatrix} \sigma_1 & & \\ & \sigma_2 & \\ & & \sigma_3 \end{bmatrix}^{-1} \begin{bmatrix} e_1^T \\ e_2^T \\ e_3^T \end{bmatrix} \qquad (20)$$

where $k_s$ is a scaling constant, and $\sigma_n$ is limited within a certain range to preserve the largest stretch. For a more detailed explanation, we recommend that you refer to Yu et al.'s work [23]. In this paper, we use the Marching Cubes algorithm based on GPU [40], and set the minimum stretch value larger than the half spacing of the grid so that thin fluid surfaces can be represented well.

## Implementations

The algorithm of this paper is implemented in the following environment: Intel i7-7700k 4.20GHz CPU, 32GB RAM, and NVIDIA GeForce GTX 1080 Ti. FLIP-based fluid solver was used as the underlying water simulation [21] and a GPU-based preconditioned conjugate gradient method was used as a numerical matrix solver to calculate fluid pressure [41]. For the FLIP grid, all momentum was stored using the Staggered Marker-and-Cell method [42] and an additional grid was used for surface reconstruction. We also used the boundary particle technique proposed by Akinci et al. for the collision of water and solid [43].

## Results

In order to analyze the proposed technique from various viewpoints, we have experimented in various scenarios.

First, as a simple test scenario, we created a scene in which a spherical liquid was dropped on a stone model (see Figs 8 ∼ 10). To represent the fluid in this scene, we set the timestep to 0.006 and used about 150,000 fluid particles. In this study, the quality of liquid sheets is analyzed by comparing naive FLIP method [21], Ando's method [4, 5], and proposed method.

Fig 8 shows the result of water simulation with naive FLIP technique. FLIP is often used to represent a variety of materials such as liquids, smoke, and deformable bodies because they

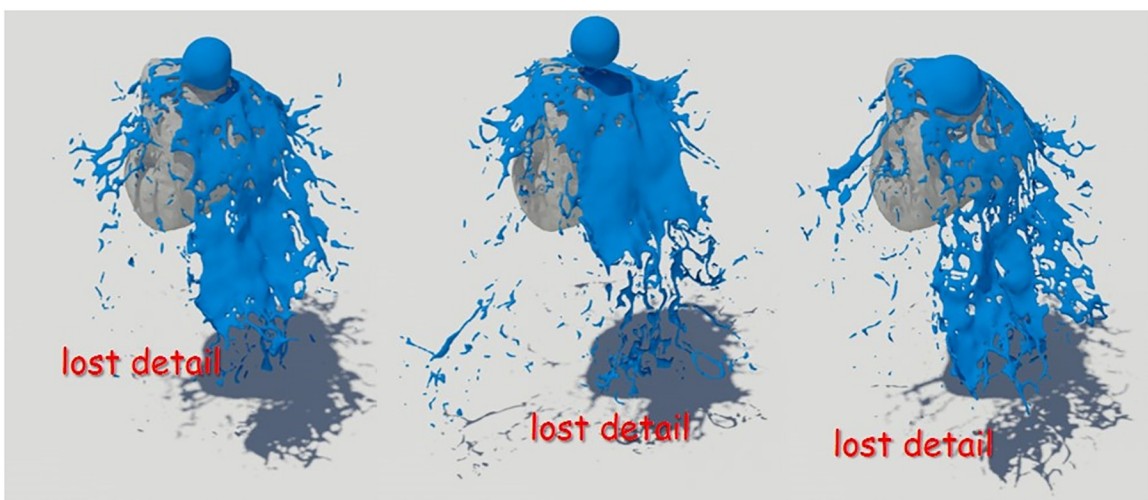

**Fig 8. Naive FLIP simulation [21].**

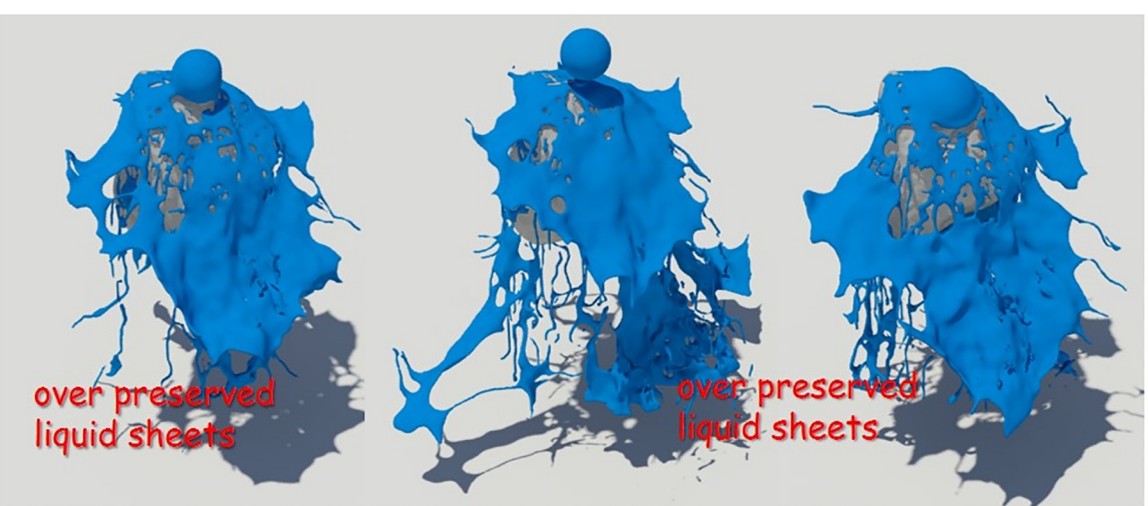

**Fig 9. Preservation fluid sheets with Ando's methods [4, 5].**

can represent dynamic fluid motion [9, 44, 45]. However, as shown in the figure, when the liquid and the solid collide, the liquid sheets, which should be expressed as a thin surface, disappear. To solve this problem, Ando saved liquid sheets by adding new fluid particles to the areas where the fluid surfaces disappeared and holes appeared [4, 5]. Although this approach shows a better fluid sheet than Naive FLIP simulation, adding new water particles to the hole area caused by surface loss leads to the problem of excessive liquid sheets (see Fig 9).

Liquid sheets should be represented in terms of preservation and breakup depending on the situation, but the approach described above focused on preservation only [4]. There have been studies to efficiently express liquid sheets by extending the framework with an adaptive and anisotropic solver, but as a result, the efficiency of computation has improved, but the quality of liquid surfaces has not improved [5]. Our method provides a more detailed representation of the preservation and breakup of liquid sheets compared to the results of previous studies (see Fig 10). The liquid sheets are preserved at the moment of collision between water

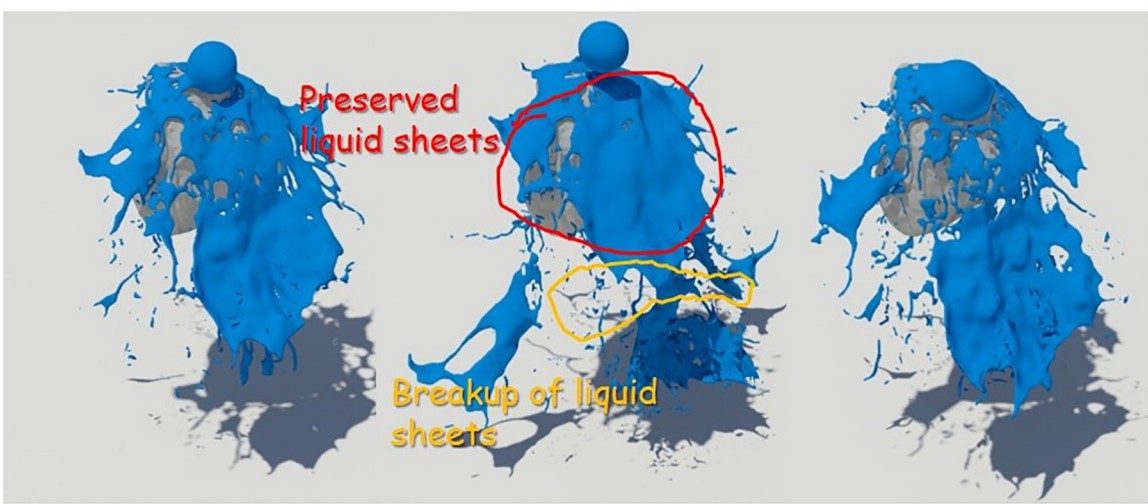

**Fig 10. Preservation fluid sheets with our method.**

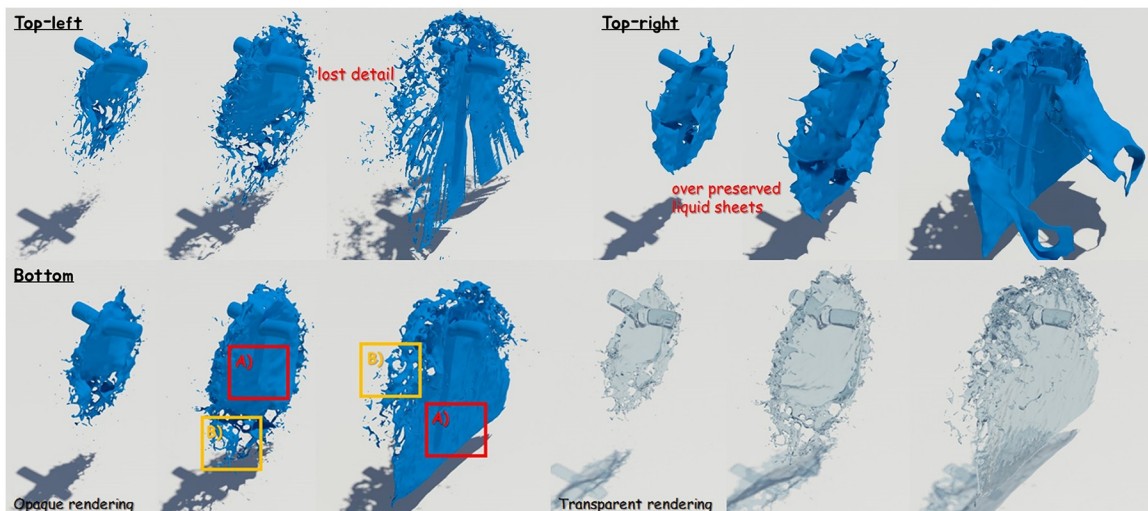

**Fig 11. Quality comparison of results: Colliding water jets (top-left: Naive FLIP simulation [21], top-right: Ando's methods [4, 5], bottom: Our method).**

and solids and we can see the fluid surfaces tearing discontinuously once they have been thinned.

Fig 11 shows a scene where two water pillars collide with each other and splash out. In this scene, the initial velocity of each water pillar is different and they collide with each other, resulting in a lot of splashes. If you calculate this scene with an existing solver, not only noise appears on the fluid surfaces, but also the shape and movement of the liquid sheet is viscous (see Fig 11-top-right). Our technique represents the splash effect caused by the collision better than the existing technique (see **B** in Fig 11-top-right), and it also expresses the features of detailed liquid sheets (see **A** in Fig 11-top-right). Conventional techniques focused only on preserving liquid sheets, so most of the splash-like features were lost (see Fig 11-top-right). This study shows visually improved results due to proper preservation and breakup of liquid sheets according to the proposed conditions (see Fig 11-bottom).

Fig 12 is another simulated result for solid interacting fluids, and this figure also shows the preservation and breakup of liquid sheets.

Previously, we described the results of expressing liquid sheets using 2D screened water particles extracted from the fixed position of the camera. Since the extraction of the screened water particles proposed in this study is dependent on the camera position, it is assumed that screened water particles will be extracted unstably if the camera rotation is very fast, which will affect the quality of the liquid sheets as a result. So we experimented with the scene where the camera rotates to see the stability and quality of the proposed method in a more dynamic environment (see Fig 13-right). Unlike our assumption, the liquid sheets are reliably expressed and no serious artifacts appeared although the camera continued to rotate (see Fig 13-right).

Simulated results show that the proposed method is better than the previous methods. In particular, we have produced stable, high-quality liquid simulation results for all scenes, rather than just showing dependence on specific scenes.

Table 1 shows the simulation environment and calculation time of each experiment scenario in this paper. It can be confirmed that not only the result quality but also the calculation efficiency is improved as compared with the previous method. In the scenes of Figs 8, 11-top-left, and 12-top-left, there was no change in the number of fluid particles because

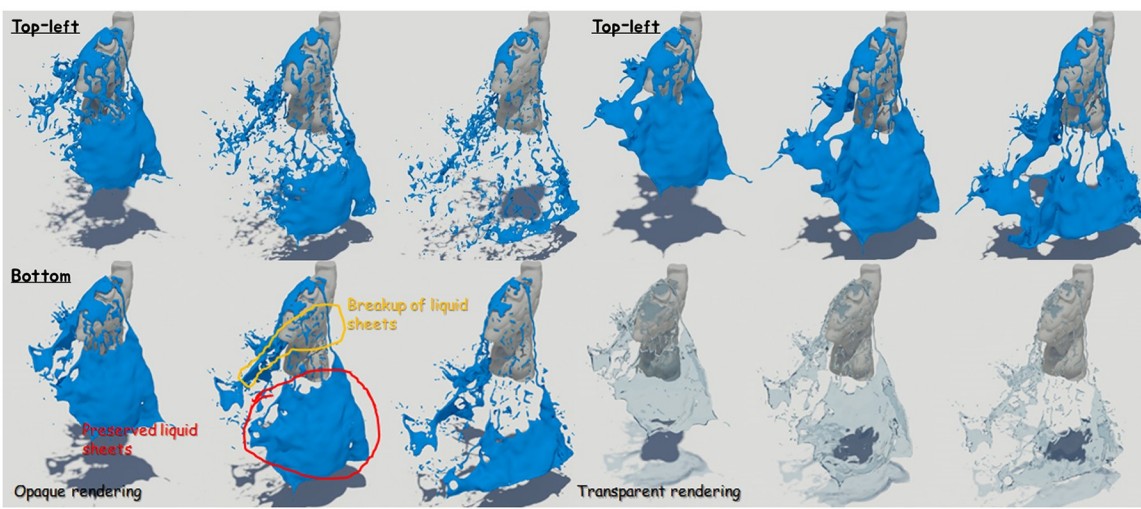

**Fig 12. Quality comparison of results: Falling water on the torso model (top-left: Naive FLIP simulation [21], top-right: Ando's methods [4, 5], bottom: Our method).**

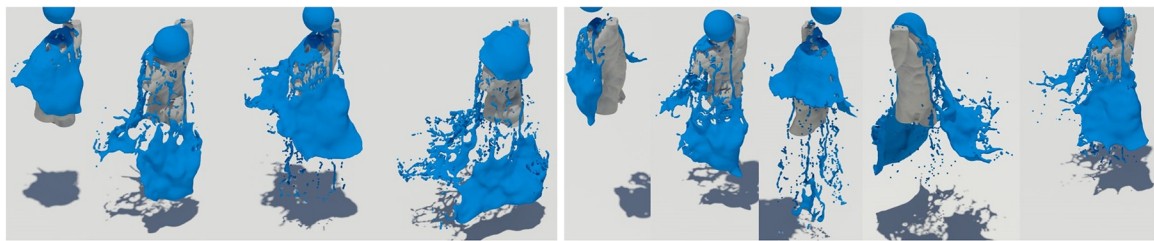

**Fig 13. Our method with screened water particles (left: With a stationary camera, right: With a rotating camera).**

**Table 1. Size of our example scenes and computational time (Num.: Number, Res.: Resolution, Comp.: Computational, projective spacing: 2.0).**

| Fig | Num. of particles | Grid res. for surface reconstruction | Comp. time (sec) with previous methods [4, 21] | Comp. time (sec) without/with screened particles | Projective space resolution |
|---|---|---|---|---|---|
| 8 | 157,047 | $256^3$ | 0.9 | – | – |
| 9 | 161,298 | $256^3$ | 1.42 | – | – |
| 10 | 157,787 | $256^3$ | – | 1.38 / 0.63 | 400×300 |
| 11-top-left | 113,910 | $256^3$ | 0.67 | – | – |
| 11-top-right | 142,122 | $256^3$ | 1.83 | – | – |
| 11-bottom | 138,703 | $256^3$ | – | 1.77 / 0.82 | 400×300 |
| 12-top-left | 154,129 | $256^3$ | 0.82 | – | – |
| 12-top-right | 161,458 | $256^3$ | 1.29 | – | – |
| 12-bottom | 159,914 | $256^3$ | – | 1.22 / 0.6 | 400×300 |
| 13-left | 160,342 | $256^3$ | – | 1.42 / 0.68 | 400×300 |
| 13-right | 162,213 | $256^3$ | – | 1.47 / 0.71 | 400×300 |

**Table 2. Simulation parameters.** We used these specific parameters to generate the example animations shown in this paper.

| Name | Description | Value |
|---|---|---|
| $\alpha_1$ | Density radius scale | 4.0 |
| $\alpha_2$ | Size of the thickness | 0.2 |
| $\alpha_3$ | Minimum insertion distance | 0.8 |
| $\alpha_4$ | Maximum insertion distance | 3.5 |
| $\alpha_5$ | Velocity radius scale | 1.0 |
| $\alpha_6$ | Maximum thin particle density | 0.2 |
| $\alpha_7$ | Maximum thin particle distance | 0.2 |
| $\tau$ | Threshold of breakup density | xx |
| $\Delta t$ | Time-step | 0.006 |

Naive FLIP simulation was used, but for the remaining results, the average number of water particles was added to the table (see Num. of particles in Table 1). The calculation time was averaged over the entire simulation process including the FLIP simulation and the surface reconstruction process (see Comp. time in Table 1). The parameters in this paper are summarized in Table 2.

In all of the simulated results of this paper, preservation and breakup of liquid sheets were implemented based on screened particles. As a result, performance is improved by about 2 times in most of the results. In actual implementation, artifacts can occur if the camera moves very fast or complexly. Therefore, it is possible to omit the screened particles process according to the user's intention and apply only the liquid sheet solver to our framework.

## Conclusions and future work

In this paper, we proposed a framework for efficiently expressing the preservation and breakup of liquid sheets, one of the representative features of liquids. For the preservation of liquid sheets we found an area where holes would be created and added new water particles there. To perform a breakup of liquid sheets, we used an average density that varies between particles when determining pairs of thin particles. The proposed method stably expresses liquid sheets in various scenes and well represented splash effects that have been lost due to over preserved liquid sheets in previous techniques. Previous approach [36, 37] are techniques for water rendering and foam simulation using a screen space approach. The difference with the proposed method is that only the water particles near the camera were selected using the depth away from the screen space to minimize the number of particles affecting the calculations during liquid sheet preservation. The shape of the particles was determined by assigning a radius to the particles, and in this process, the accuracy of the depth value varies depending on the size of the radius. Our paper does not calculate the exact depth map, but rather selects screened water particles, which yields sufficient computational efficiency in pruning unwanted particles. In addition, when projecting 3D water particles advected by 3D FLIP (Fluid-implicit particle) used as an underlying water solver to 2D screen space, only x and y information is projected. Since they have similar characteristics and motions to those obtained by solving the Navier-Stokes equation, they are approximated by selecting screened water particles. Of course, there are some differences from the exact Navier-stokes equation movement.

In the future, we plan to train the liquid sheets with a neural network instead of calculating the movement of the particles every frame, so as to predict the liquid detail efficiently.

## Supporting information

**S1 Video. Supplementary result data.** Related to Fig 1.
(AVI)

## Acknowledgments

This work was supported in part by the Basic Science Research Program through the National Research Foundation of Korea (NRF) funded by the Ministry of Science, ICT & Future Planning (2017R1C1B5074984), and in part by a Hallym University Research Fund (HRF-201909-018).

## Author Contributions

**Conceptualization:** Jong-Hyun Kim, Jung Lee.

**Data curation:** Jong-Hyun Kim.

**Formal analysis:** Jong-Hyun Kim, Jung Lee.

**Investigation:** Jong-Hyun Kim.

**Methodology:** Jong-Hyun Kim, Jung Lee.

**Project administration:** Jong-Hyun Kim.

**Resources:** Jung Lee.

**Software:** Jong-Hyun Kim.

**Supervision:** Jong-Hyun Kim.

**Validation:** Jong-Hyun Kim, Jung Lee.

**Visualization:** Jong-Hyun Kim.

**Writing – original draft:** Jong-Hyun Kim, Jung Lee.

**Writing – review & editing:** Jong-Hyun Kim, Jung Lee.

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
