## [Decision Letter · Decision Letter 0]

30 Oct 2019

PONE-D-19-14945

Efficient Preservation and Breakup of Liquid Sheets Using Screen-Projected Particles

PLOS ONE

Dear Jung Lee,

Thank you for submitting your manuscript to PLOS ONE. After careful consideration, we feel that it has merit but does not fully meet PLOS ONE’s publication criteria as it currently stands. Therefore, we invite you to submit a revised version of the manuscript that addresses the points raised during the review process.

We would appreciate receiving your revised manuscript by Dec 14 2019 11:59PM. To enhance the reproducibility of your results, we recommend that if applicable you deposit your laboratory protocols in protocols.io, where a protocol can be assigned its own identifier (DOI) such that it can be cited independently in the future. For instructions see: http://journals.plos.org/plosone/s/submission-guidelines#loc-laboratory-protocols

We look forward to receiving your revised manuscript.

Kind regards,

Ajaya Bhattarai

Academic Editor

PLOS ONE

Journal Requirements:

3. Please re-upload the Video file as it is currently not accessible. Please also upload this as a "supporting information" file type.

 "The funders had no role in study design, data collection and analysis, decision to publish, or preparation of the manuscript.".

** Please provide an amended Funding Statement that declares *all* the funding or sources of support received during this specific study ** (whether external or internal to your organization) as detailed online in our guide for authors at http://journals.plos.org/plosone/s/submit-now.  

Please state what role the funders took in the study.  If any authors received a salary from any of your funders, please state which authors and which funder. If the funders had no role, please state: "The funders had no role in study design, data collection and analysis, decision to publish, or preparation of the manuscript."

Reviewers' comments:

Reviewer's Responses to Questions

**Comments to the Author**

1. Is the manuscript technically sound, and do the data support the conclusions?

Reviewer #1: Yes

Reviewer #2: Yes

2. Has the statistical analysis been performed appropriately and rigorously? 

Reviewer #1: N/A

Reviewer #2: N/A

3. Have the authors made all data underlying the findings in their manuscript fully available?

Reviewer #1: No

Reviewer #2: Yes

4. Is the manuscript presented in an intelligible fashion and written in standard English?

Reviewer #1: Yes

Reviewer #2: Yes

5. Review Comments to the Author

Reviewer #1: -------------------------------------

See attachment.

Reviewer #2: The main idea and implementation details sounds solid and contributes the thin sheet augmentation techniques for the particle based fluids as they introduce new break-up criteria, which is simple and effective.

For the evaluations, It would be great to see more experiment results and comparisons on the following cases

- using full 3d particles to select candidates vs. screen particles to select candidates, to see difference in visual fidelity.

- how the algorithm works for a rapid camera motions, would be it be robust without exhibiting abrupt and non-natural artifacts on the thin sheet constructions.

6. PLOS authors have the option to publish the peer review history of their article (what does this mean?). If published, this will include your full peer review and any attached files.

Reviewer #1: No

Reviewer #2: No

---

## [Author Response · Author response to Decision Letter 0]

11 Dec 2019

[Download] Attach Files-Revision_Letter.docx

---

## [Editor Report · Decision Letter 1]

23 Dec 2019

Efficient Preservation and Breakup of Liquid Sheets Using Screen-Projected Particles

PONE-D-19-14945R1

Dear Dr. Jung Lee,

We are pleased to inform you that your manuscript has been judged scientifically suitable for publication and will be formally accepted for publication once it complies with all outstanding technical requirements.

With kind regards,

Ajaya Bhattarai

Academic Editor

PLOS ONE
---

## [Editor Report · Acceptance letter]

24 Jan 2020

PONE-D-19-14945R1 

Efficient Preservation and Breakup of Liquid Sheets Using Screen-Projected Particles 

Dear Dr. Lee:

I am pleased to inform you that your manuscript has been deemed suitable for publication in PLOS ONE. Congratulations! Your manuscript is now with our production department. 

With kind regards,

on behalf of

Dr. Ajaya Bhattarai 

Academic Editor

PLOS ONE